Jumping performance in the highly aquatic frog, Xenopus tropicalis: sex-specific relationships between morphology and performance

Herrel Anthony 1 2 anthony.herrel@mnhn.fr
Vasilopoulou-Kampitsi Menelia 1
Bonneaud Camille 3
1 UMR 7179, CNRS/MNHN, Département d’Ecologie et de Gestion de la Biodiversité , Paris Cedex , France
2 Ghent University, Evolutionary Morphology of Vertebrates , Gent , Belgium
3 Centre for Ecology & Conservation, College of Life and Environmental Sciences, University of Exeter , Penryn, Cornwall , UK
Hutchinson John
Electronic publication date: 2014 Nov 4
Publication date: 2014
Volume: 2
Electronic Location ID: e661
Received 2014 Sep 4; Accepted 2014 Oct 20
Copyright: © 2014 Herrel et al.
Copyright year: 2014
Copyright holder: Herrel et al.
License: This is an open access article distributed under the terms of the Creative Commons Attribution License, which permits unrestricted use, distribution, reproduction and adaptation in any medium and for any purpose provided that it is properly attributed. For attribution, the original author(s), title, publication source (PeerJ) and either DOI or URL of the article must be cited.
License URL: https://creativecommons.org/licenses/by/4.0/

Keywords: Locomotion, Trade-off, Jumping, Frog, Sexual dimorphism

Funding: ANR MOBIGEN ANR-09-PEXT-003 MNHN ATM Marie Curie reintegration grant FP7-PEOPLE-IRG-2008 #239257 Financial support was provided by ANR MOBIGEN to AH and CB (ANR-09-PEXT-003), a MNHN ATM grant of the programme ‘Biodiversité actuelle et fossile’ to AH, and a Marie Curie reintegration grant to CB (FP7-PEOPLE-IRG-2008 #239257). The funders had no role in study design, data collection and analysis, decision to publish, or preparation of the manuscript.

==============================
Frogs are characterized by a morphology that has been suggested to be related to their unique jumping specialization. Yet, the functional demands associated with jumping and swimming may not be that different as suggested by studies with semi-aquatic frogs. Here, we explore whether features previously identified as indicative of good burst swimming performance also predict jumping performance in a highly aquatic frog, Xenopus tropicalis. Moreover, we test whether the morphological determinants of jumping performance are similar in the two sexes and whether jumping performance differs in the two sexes. Finally we test whether jumping capacity is positively associated with burst swimming and terrestrial endurance capacity in both sexes. Our results show sex-specific differences in jumping performance when correcting for differences in body size. Moreover, the features determining jumping performance are different in the two sexes. Finally, the relationships between different performance traits are sex-dependent as well with females, but not males, showing a trade-off between peak jumping force and the time jumped to exhaustion. This suggests that different selective pressures operate on the two sexes, with females being subjected to constraints on locomotion due to their greater body mass and investment in reproductive capacity. In contrast, males appear to invest more in locomotor capacity giving them higher performance for a given body size compared to females.

Introduction

Frogs are characterized by a morphology that includes elongated ilia, a shortening of the presacral vertebral series, the fusion of the caudal vertebral elements into an urostyle, and the presence of mobile ilio-sacral and sacro-urostylic joints. These features have been suggested to be related to their unique jumping specialization that originated early-on in their evolutionary history (Shubin & Jenkins, 1995). The mobility of the ilio-sacral and sacro-urostylic joints is thought to be a key feature for jumping as it allows an extension of the body and thus the transfer of propulsive forces by the coccygeo-iliacus muscles. In specialized aquatic frogs, on the other hand, the ilio-scaral joint is also mobile but highly modified. In Xenopus frogs this joint allows sliding of the ilia along the sacral vertebra which is thought to aid in propulsion (Palmer, 1960; Videler & Jorna, 1985). Given the radical differences in the ilio-sacral anatomy in specialized swimmers versus jumpers, specialized aquatic frogs are consequently thought to be rather poor jumpers (Olson & Marsh, 1998). This idea is also supported by broad comparative analyses of jumping performance in frogs where semi-aquatic frogs typically show intermediate levels of performance (Zug, 1972; Zug, 1978; Gomes et al., 2009).

However, despite the anatomical differences in the ilio-sacral joint between specialized aquatic species and terrestrial jumpers, the functional demands associated with jumping and swimming may not be that different. For example, in semi-aquatic frogs, no trade-offs between swimming and jumping capacity could be demonstrated, suggesting that no direct design conflict is present in animals performing both types of locomotion on a daily basis (Nauwelaerts, Ramsay & Aerts, 2007). Moreover, different anatomical traits explained swimming versus jumping performance in these semi-aquatic frogs which may explain the absence of a correlation between the two performance measures. However, given the role of elastic energy storage in the plantaris longus tendon in determining jumping performance (Astley & Roberts, 2011), aquatic frogs with shorter plantaris tendons may need to invest in greater absolute muscle cross sectional areas to achieve high levels of jumping performance. As such, the traits that determine jumping and swimming may be more similar in specialized aquatic frogs compared to semi-aquatic species.

Here, we build upon an existing data set on locomotor performance in the specialized aquatic frog, Xenopus (Silurana) tropicalis (Herrel & Bonneaud, 2012a; Herrel & Bonneaud, 2012b), to explore whether trade-offs exist between swimming and jumping and whether similar morphological traits are the best predictors of the two performance traits. Based on the results of Nauwelaerts and co-workers (2007), we predict that different anatomical features will determine the two performance traits resulting in the absence of a correlation between both. Moreover, we predict that jumping performance should be correlated to overall exertion capacity, as the energetics of jumping likely limit the number of jumps rather than the overall distance jumped per se. Thus the distance jumped to exhaustion should thus be positively correlated to jumping performance. However, a greater proportion of fast glycolytic muscle fibers in the muscles responsible for jumping may cause animals to fatigue quicker (James, Navas & Herrel, 2007) and as such may in theory cause a trade-off between jump force and the distance jumped to exhaustion.

We also test whether jumping performance differs between the two sexes and whether the morphological determinants of jumping performance are similar in the two sexes. Given that X. tropicalis frogs are dimorphic in body mass and limb dimensions, with females being heavier but males having relatively longer limbs (Herrel & Bonneaud, 2012a), we predict that males are better jumpers for a given body size or hind limb length than females. Indeed, females invest relatively more energy in reproductive output than males in many animals, including frogs (Shine, 1979). Thus, females are often bigger and heavier than males in species where males do not engage in male–male combat (Schauble, 2004). Moreover, most of the extra body mass is involved in reproductive output, thus increasing the load relative to the available muscle mass and cross sectional area. Finally, given that the sexes are known to differ in swimming performance and endurance capacity (Herrel & Bonneaud, 2012a), we also explore sex-specific correlations between the different locomotor performance traits.

Materials and Methods

Animals

Xenopus tropicalis were caught in the wild in December 2009 in Cameroon brought back to France and housed at the Station d’Ecologie Experimentale du CNRS at Moulis. Animals were housed in groups of 8–10 individuals in aquaria (60 × 30 × 30 cm) with the temperature set at 24 °C which is assumed to be close to the preferred and optimal temperature of Xenopus frogs (Casterlin & Reynolds, 1980; Miller, 1982), and similar to water temperatures measured in the field in ponds where the animals were caught (22–26 °C; Careau et al., 2014). Frogs were fed every other day with beef heart, earthworms, or mosquito larvae ad libitum. All individuals were given one month to recover and were then pit-tagged (NONATEC) before the onset of the experiments allowing unambiguous identification. A total of 125 individuals were included in the performance testing. All experiments were approved by the Institutional ethics committee at the MNHN (#68-25).

Morphometrics

All animals (N = 125; 56 males and 69 females) were weighed (Ohaus, precision ±0.01 g) and measured using digital calipers (Mitutoyo, ±0.01 mm). The following body dimensions were quantified: body length as the straight-line distance from the cloaca to the tip of the snout, the length of the femur, the tibia, the foot, the longest toe, the ilium and the width across the top of the two ilia (see Herrel et al., 2012).

Performance

All performance traits were measured at 24 °C. Before the onset of performance measurements, animals were placed for one hour in an incubator set at 24 °C in individual containers with some water. All performance measurements were repeated three times over the course of one day for each individual with an inter-trial interval of at least one hour during which animals were returned to the incubator and allowed to rest. At the end of the performance trials animals were weighed, their pit tag numbers recorded and they were returned to their home aquaria and fed. Animals were given at least one week rest between the different performance measures.

Data on maximal exertion capacity and swimming performance were taken from Herrel & Bonneaud (2012a). Repeatabilities of these traits are listed in Careau et al. (2014). In brief, maximal exertion capacity was measured by chasing each individual down a 3 m long circular track until exhaustion, indicated by unwillingness to move any further when touched and the lack of a righting response (inability to turn when animals are placed on their backs). Burst performance capacity was quantified by measuring maximal instantaneous swimming speed and acceleration of animals filmed with a Redlake MotionPro high speed camera set at 500 Hz.

Maximal jump forces were measured using a piezo-electric force platform (Kistler Squirrel force plate, 0.1 N). The force platform (20 by 10 cm) was connected to a charge amplifier (Kistler Charge Amplifier type 9865) and forces were recorded at 500 Hz, transferred to the computer, and recorded using Bioware software (Kistler). Frogs were placed on the force plate, allowed to rest for a few seconds and then induced to jump by unexpectedly clapping our hands behind the frogs. This elicited maximal escape responses from the individuals causing them to jump as far as possible away from the observer. Frogs were caught and placed back on the force plate as many times as possible during the 60 s recording time. Three jump sessions with three to five jumps each on average were recorded and the single most forceful jump was retained out of all jumps recorded and used for further analyses. Forces in X, Y and Z-directions were extracted (Fig. 1) using the Kistler Bioware software and the total resultant force (Fres: vector sum of the X, Y and Z forces) as well as the force in the vertical (Z; Fz) and horizontal (X + Y; FXY) planes were calculated. Note that as the position of the frog on the force plate was random (i.e., as preferred by the animal), X- and Y- forces do not represent the fore-aft and medio-lateral forces per se. Thus in one jump the X may be aligned with the direction of jumping and in another the Y or neither. Jump forces were repeatable across trials (intra-class correlation coefficients Fz: r = 0.826, P < 0.001; FXY:r = 0.637, P < 0.001; Fres:r = 0.814, P < 0.001).

Figure 1 Example force trace from a female X. tropicalis jumping.

Indicated are the Z (vertical), X (short axis of the force plate) and Y (long axis of the force plate) forces. Note that the animal is not always positioned in line with the long axis of the force plate, and that horizontal forces cannot be interpreted in terms of fore-aft or medio-lateral forces. When the animal is placed on the force plate the Z-force increases as a result of the weight of the animal as indicated in the figure. Jumping is characterized by a rapid increase in the vertical, as well as in the horizontal forces.

Figure 2 Scatter plots illustrating the relationships between morphology and the peak resultant force for female (A) and male (B) frogs.

While hind limb length is the best predictor of jump force in females, the length of the ilium is the best predictor in males (r = 0.467; P < 0.001; see Table 1). Thus females with longer legs and males with longer ilia are better jumpers (r = 0.717; P < 0.001; see Table 1). Each symbol represents the single best jump for an individual. Open symbols represent females, filled symbols represent males.

Figure 3 Scatter plot illustrating the differences in the resultant jump force for a given body mass.

Note that males (intercept = −0.78; slope = 0.66; R2 = 0.15; P = 0.003) are better jumpers than females (intercept = −1.05; slope = 0.88; R2 = 0.50; P < 0.001) for their size (Table 1). Each symbol represents the single best jump for an individual. Open symbols represent females, filled symbols represent males.

Figure 4 Scatter plots illustrating the relationships between jumping force and endurance capacity.

Whereas the distance jumped until exhaustion is positively correlated with jump force in both sexes, the time jumped until exhaustion is positively correlated in males but negatively correlated to peak jump force in females (see Table 2). Each symbol represents the single best jump for an individual. Open symbols represent females, filled symbols represent males.

Analyses

All data were Log10-transformed before analyses to fulfill assumptions of normality and homoscedascity. First, we ran analyses of variance to test for differences in jump force between the two sexes. Given that females are larger and heavier than males we also ran analyses of covariance with body mass and hind limb length as covariates. Next, we ran stepwise multiple regressions to explore which morphological traits (SVL, mass, limb segment lengths, total hind limb length) determined variation in jumping force for all individuals as well as for both sexes separately (Fig. 2). Finally, we ran Pearson correlations between all morphological traits and jump forces (Table 1) and between the forces the different performance traits (Table 2) to test for the presence of potential trade-offs between performance traits, again for the entire data set as well as for both sexes separately. All analyses were performed using SPSS v. 15.0.

Table 1 Pearson correlations between jump forces and morphological traits.

The single highest correlation is indicated in bold.

	SVL
(mm)	Mass
(g)	Femur
(mm)	Tibia
(mm)	Foot
(mm)	Toe
(mm)	Hind limb
(mm)	Ilium length
(mm)	Ilium width
(mm)	
All individuals (N = 125)	
Resultant force (N)	r = 0.593
P < 0.001	r = 0.594
P < 0.001	r = 0.590
P < 0.001	r = 0.613
P < 0.001	r = 0.605
P < 0.001	r = 0.584
P < 0.001	r = 0.616
P < 0.001	r = 0.586
P < 0.001	r = 0.501
P < 0.001	
Vertical force (N)	r = 0.609
P < 0.001	r = 0.599
P < 0.001	r = 0.593
P < 0.001	r = 0.629
P < 0.001	r = 0.602
P < 0.001	r = 0.545
P < 0.001	r = 0.620
P < 0.001	r = 0.592
P < 0.001	r = 0.519
P < 0.001	
Horizontal force (N)	r = 0.466
P < 0.001	r = 0.493
P < 0.001	r = 0.469
P < 0.001	r = 0.499
P < 0.001	r = 0.508
P < 0.001	r = 0.442
P < 0.001	r = 0.501
P < 0.001	r = 0.491
P < 0.001	r = 0.448
P < 0.001	
Males (N = 56)	
Resultant force (N)	r = 0.377
P = 0.004	r = 0.389
P = 0.003	r = 0.220
P = 0.103	r = 0.333
P = 0.012	r = 0.378
P = 0.004	r = 0.114
P = 0.404	r = 0.284
P = 0.034	r = 0.467
P < 0.001	r = 0.306
P = 0.022	
Vertical force (N)	r = 0.459
P < 0.001	r = 0.431
P = 0.001	r = 0.268
P = 0.046	r = 0.421
P = 0.001	r = 0.395
P = 0.003	r = 0.142
P = 0.295	r = 0.337
P = 0.011	r = 0.497
P < 0.001	r = 0.276
P = 0.039	
Horizontal force (N)	r = 0.125
P = 359	r = 0.218
P = 0.106	r = 0.225
P = 0.095	r = 0.219
P = 0.104	r = 0.251
P = 0.062	r = 0.147
P = 0.279	r = 0.232
P = 0.086	r = 0.209
P = 0.122	r = 0.391
P = 0.003	
Females (N = 69)	
Resultant force (N)	r = 0.694
P < 0.001	r = 0.707
P < 0.001	r = 0.693
P < 0.001	r = 0.710
P < 0.001	r = 0.690
P < 0.001	r = 0.674
P < 0.001	r = 0.717
P < 0.001	r = 0.673
P < 0.001	r = 0.601
P < 0.001	
Vertical force (N)	r = 0.711
P < 0.001	r = 0.721
P < 0.001	r = 0.696
P < 0.001	r = 0.720
P < 0.001	r = 0.692
P < 0.001	r = 0.671
P < 0.001	r = 0.720
P < 0.001	r = 0.688
P < 0.001	r = 0.648
P < 0.001	
Horizontal force (N)	r = 0.536
P < 0.001	r = 0.552
P < 0.001	r = 0.520
P < 0.001	r = 0.563
P < 0.001	r = 0.569
P < 0.001	r = 0.518
P < 0.001	r = 0.560
P < 0.001	r = 0.549
P < 0.001	r = 0.443
P < 0.001	
Notes.

SVL snout-vent length

Table 2 Pearson correlations among performance traits.

Bolded values represent significant correlations.

	Max. speed (m s−1)	Max acceleration (m s−2)	Time to exhaustion (s)	Distance to exhaustion (cm)	
All individuals (N = 125)	
Resultant force (N)	r = 0.056
P = 0.538	r = −0.008
P = 0.925	r = −0.112
P = 0.215	r = 0.269
P = 0.003	
Vertical force (N)	r = 0.017
P = 0.849	r = −0.053
P = 0.556	r = −0.107
P = 0.241	r = 0.265
P = 0.003	
Horizontal force (N)	r = 0.178
P = 0.047	r = 0.003
P = 0.977	r = −0.028
P = 0.760	r = 0.241
P = 0.007	
Males (N = 56)	
Resultant force (N)	r = −0.220
P = 0.104	r = −0.125
P = 0.357	r = 0.188
P = 0.170	r = 0.320
P = 0.017	
Vertical force (N)	r = −0.188
P = 0.165	r = −0.181
P = 0.183	r = 0.156
P = 0.256	r = 0.346
P = 0.010	
Horizontal force (N)	r = 0.039
P = 0.776	r = −0.005
P = 0.969	r = 0.192
P = 0.161	r = 0.161
P = 0.241	
Females (N = 69)	
Resultant force (N)	r = 0.206
P = 0.090	r = 0.052
P = 0.673	r = −0.263
P = 0.030	r = 0.273
P = 0.024	
Vertical force (N)	r = 0.135
P = 0.270	r = 0.016
P = 0.893	r = −0.243
P = 0.046	r = 0.246
P = 0.043	
Horizontal force (N)	r = 0.242
P = 0.045	r = −0.028
P = 0.817	r = −0.147
P = 0.230	r = 0.344
P = 0.004	

Results

Peak forces ranged from 0.113 N in an animal of 35.5 mm and 4.72 g to 1.69 N resultant force recorded for an animal of 48.5 mm and 10.7 g. Thus frogs produced between ten and 20 times their own body mass in jump force. The mean resultant jump force was 0.53 ± 0.26 N for an average body length of 38.74 ± 6.09 mm and an average mass of 6.37 ± 2.85 g. Peak vertical forces (0.076–1.52 N) were greater than forces in the horizontal plane (0.02–1.22 N).

Sexual dimorphism in jump forces

Analyses of variance testing for differences in jump force between males and females detected no differences in peak resultant force (F1,123 = 0.02; P = 0.89), peak vertical force (F1,123 = 0.017; P = 0.90), nor peak horizontal force (F1,123 = 3.46; P = 0.07). However, when taking into account hind limb length significant differences in peak resultant force (F1,122 = 4.14; P = 0.044) and peak vertical force (F1,122 = 5.91; P = 0.016) were detected with males producing higher forces for a given hind limb length (Fig. 3). Similarly, when using body mass as a covariate, significant differences in peak resultant force (F1,122 = 13.08; P < 0.001) and peak vertical force (F1,122 = 16.94; P < 0.001) were observed with males again showing higher forces than females for a given body mass.

Morphology∼performance correlates

Raw correlations between morphological variables and jump forces are in Table 1. A stepwise regression analyses with the peak resultant force as the dependent variable and the different morphological traits as independents for the whole data set resulted in a highly significant model with hind limb length as the only variable (R2 = 0.38; P < 0.001). A similar analyses run on the data for males only retained a model with ilium length and the length of the longest toe as explicatory variables (R2 = 0.29; P < 0.001) (Fig. 2). Partial standardized regression coefficients indicated that whereas ilium length was positively associated with peak jump force (β = 0.70), toe length was negatively correlated (β = −0.35) with peak jump force. For females a model with only hind limb length was retained (R2 = 0.51; P < 0.001) (Fig. 2). Peak vertical forces (Z) were best predicted by tibia length (R2 = 0.40; P < 0.001) for the entire data set, by ilium length and toe length for males (R2 = 0.31; P < 0.001; ilium length β = 0.72; toe length β = −0.34), and by body mass in females (R2 = 0.52; P < 0.001). Peak force in the horizontal plane, on the other hand, were best predicted by foot length for the overall data set (R2 = 0.26; P < 0.001), by ilium width in males (R2 = 0.15; P = 0.003), and by foot length in females (R2 = 0.32; P < 0.001).

Performance relationships

When considering the overall data set significant positive relationships were detected between peak resultant force, peak vertical force and peak horizontal force and the distance jumped to exhaustion (Table 2). Moreover, the peak horizontal forces were positively correlated with the peak swimming velocity (Table 2). When analyzing data for males only, the peak resultant force and the peak vertical force were correlated to the distance jumped to exhaustion (Table 2). In females, the peak resultant force, the peak vertical force and the peak horizontal force were positively correlated to the distance jumped to exhaustion (Table 2, Fig. 4). Moreover, the peak resultant and vertical forces were negatively correlated to the time jumped until exhaustion (Table 2). Finally, the peak horizontal force was also positively correlated to the peak swimming velocity (Table 2).

Discussion

Sexual dimorphism in jump forces

Although absolute jump forces did not differ between the two sexes, a sexual dimorphism in performance became apparent when correcting for known differences between the sexes in hind limb length or body mass (Herrel et al., 2012). Indeed, when taking into account differences in morphology, males showed higher jump forces than females. This result mimics the observation that males also have a higher endurance capacity in terms of the distance jumped to exhaustion compared to females (Herrel et al., 2012). Given that peak jump forces are correlated to the distance jumped to exhaustion, this makes intuitive sense. Although males are smaller than females, they have relatively longer hind limbs than females (Herrel et al., 2012). Given that hind limb length is correlated to jump force, this explains at least in part while males have a greater relative jumping performance. However, even when correcting for differences in hind limb length, males still have a greater jump forces than females. This suggests intrinsic sex-differences, with males investing more in skeletal muscle tissue and females more in reproductive output. This is confirmed by our analyses indicating that males also have a greater jumping performance when correcting for differences in body mass. Males of X. tropicalis typically show high levels of exploration behavior (Videlier et al., in press) and thus, locomotor capacity may be under strong selection. Whether females explore their environment less than males, however, remains to be tested. If females should indeed be more stationary, then this would explain their lower investment in terrestrial locomotor performance in benefit of reproductive output.

Morphological determinants of jumping performance

Our analyses suggest that peak resultant force is principally determined by hind limb with animals with longer hind limbs generating larger forces in correspondence with previous studies (Nauwelaerts, Ramsay & Aerts, 2007). While hind limb length was also the best predictor of the peak resultant force for females, ilium length and the length of the longest toe were retained for males. Males with longer ilia, yet shorter longest toes produced greater forces. That the length of the ilium was retained in the model suggest that a caudopelvic mechanism (Jenkins & Shubin, 1998) may be operational in aquatic frogs like X. tropicalis. Although previous authors have suggested that in aquatic frogs, pelvic sliding may play an important role in augmenting propulsion during swimming (Videler & Jorna, 1985), our data suggest that this may play a more important role during jumping than swimming. As the coccygeo-sacralis, the coccygeo-iliacus, and the longissumus dorsi muscles all span the joint between the vertebral column and the pelvic girdle, they may help extend the iliosacral joint. This, in turn, may transfer force to the other joints of the limb and help increase the peak resultant forces (e.g., Aerts, 1998).

Peak vertical forces, in contrast, were best predicted by tibia length for the entire data set. However, in males, ilium length and toe length were the best predictors, while in females, body mass was the best predictor of peak vertical force. The peak forces in the horizontal plane, on the other hand, were best predicted by foot length for the overall data set, by ilium width in males, and by foot length in females. In contrast to the resultant forces, peak forces in the vertical and horizontal planes are determined by different anatomical features when considering males, females and the overall data set. Although this result may be indicative of real differences between the sexes in the morphological traits that determine jumping performance, further experiments involving simultaneous high-speed video and force recordings are needed to better understand why this is the case. For example, differences in jump angle or the contribution of the forelimb in changing body posture before the onset of the jump may all play a role in determining what variables determine peak force. Clearly, only part of the variation in jump force was explained by external morphology with between 15 and 52% of the overall variation in jump force explained. Thus it would be of interest to explore differences in the limb muscles and their physiological properties as it is known that these play an important role in determining jumping ability in a variety of vertebrate species, including frogs (e.g., Aerts, 1998; Harris & Steudel, 2002; Toro, Herrel & Irschick, 2004; James, Navas & Herrel, 2007; Nauwelaerts, Ramsay & Aerts, 2007). Specifically, differences in muscle cross sectional area or in muscle fiber type may be important in explaining inter-individual as well as inter-sexual differences in jumping capacity (James et al., 2005). For example, significant differences in muscle size and physiology have been noted between the sexes in the forearm muscles in explosive breeders where males compete to gain hold of females (Navas & James, 2007). Although such mating competition has never been described for X. tropicalis sex-dependent differences in muscle physiology may exist and would be of interest to explore further.

Trade-offs in locomotor performance

In the overall data set, no performance trade-offs were detected. Rather, forces were positively correlated to the distance jumped to exhaustion. This result is expected if the energetics of jumping likely limit the number of jumps rather than the actual jump distance given the potential role for energy storage and amplification in jumping in frogs (Lutz & Rome, 1994; Astley & Roberts, 2011). Thus a more forceful jump will result in a greater jump distance if jump angle remains unaffected (Marsh, 1994) and thus jump forces should be correlated to total distance jumped until exhaustion. Interestingly, peak horizontal forces were also correlated to peak swimming velocity. Given that swimming involves limb extension in the horizontal plane (Gal & Blake, 1988; Nauwelaerts, Stamhuis & Aerts, 2005; Richards, 2010), the ability of a frog to generate greater forces in the horizontal plane may indeed be logically related to its peak swimming velocity. Previous authors have found no correlation between swimming and jumping ability in a semi-aquatic frog (Nauwelaerts, Ramsay & Aerts, 2007). However, these authors investigated only the peak resultant force and did not investigate whether peak forces in the horizontal plane were correlated with peak swimming velocity (Nauwelaerts, Ramsay & Aerts, 2007). Indeed, peak resultant force and swimming speed were not correlated in our dataset either.

Data for males were similar to the results for all individuals combined with the exception that no correlation between peak horizontal force and swimming velocity was observed. In females, however, negative relationships between the peak resultant and peak vertical force and the time jumped until exhaustion were observed in addition to the correlation observed for the overall data set. Thus, in females peak jump force trades-off with the time jumped to exhaustion. This suggests that females that produce greater jump forces get tired sooner and stop jumping earlier. As peak resultant and vertical forces are correlated to overall hind limb length and body mass respectively, this suggests that larger females get tired sooner, despite the fact that they can generate larger forces and thus jump further. This is likely due to the differential scaling of force relation to body mass. Indeed whereas body mass increases with hind limb length to the third power, force only increases to the second power (Hill, 1950). Thus larger females have to move a relative larger weight against gravity. In addition, as larger animals tend to rely less on elastic energy storage for jumping (James, Navas & Herrel, 2007), this suggests a greater energetic cost in animals producing higher forces causing them to tire sooner.

In summary, our results show sex-specific differences in jumping performance when correcting for differences in body size. Moreover, the features determining jumping performance are different in the two sexes. Finally, the relationships between different performance traits are sex-dependent with peak jumping force trading-off with the time jumped to exhaustion in females, but not in males. This suggests that different selective pressures may be operating on the two sexes. Females are subjected to constraints on locomotion due to their greater body mass. Indeed, investment in reproductive output (i.e., large egg mass) causes females to be relatively heavier than males. Males, despite being smaller, are capable of generating the same jump forces as females suggesting that they may have larger limb muscles and/or more fast glycolytic muscle fibers. Thus, males appear to invest more energy in traits related to locomotion giving them a higher locomotor performance for a given body size compared to females. This may be of benefit to males as they are known to disperse in periods of heavy rain to seek mates (Rödel, 2000). However, additional studies are needed to better understand the selective pressures operating on males and females in wild populations and how these may influence the evolution of locomotor traits in these frogs.

Supplemental Information

Supplemental Information 1 jump forces

File providing raw peak jump force data for each individual. Provided are the resultant forces, the peak vertical forces and the peak horizontal forces.

Click here for additional data file.

We would like to thank Legrand Nono Gonwouo and Erik Fokam for their valuable help in the field and O Calvez, J Rodriguez, N Bonneau, and P Provini for helping to take care of the frogs, and M Antoine for logistical help.

Additional Information and Declarations

Competing Interests

Author Contributions

Animal Ethics

The authors declare there are no competing interests.

Anthony Herrel conceived and designed the experiments, performed the experiments, analyzed the data, contributed reagents/materials/analysis tools, wrote the paper, prepared figures and/or tables, reviewed drafts of the paper.

Menelia Vasilopoulou-Kampitsi conceived and designed the experiments, performed the experiments, analyzed the data, wrote the paper, reviewed drafts of the paper.

Camille Bonneaud conceived and designed the experiments, contributed reagents/materials/analysis tools, wrote the paper, reviewed drafts of the paper.

The following information was supplied relating to ethical approvals (i.e., approving body and any reference numbers):

All experiments were approved by the institutional ethics committee at the MNHN (#68-025; Comite Cuvier).

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
