# Peer review of "Jumping performance in the highly aquatic frog, Xenopus tropicalis: sex-specific relationships between morphology and performance"

_PeerJ, doi:10.7717/peerj.661_

## Round 0.1 · original submission · Major Revisions

· Academic Editor

Major Revisions

The two reviewers have provided very constructive and insightful criticisms of the paper that would improve it and make it acceptable.

Please address all of their points individually in a detailed Response document and remember to include a copy of your text with Tracked Changes to ease the review process. Reviewers will need to take a 2nd look at the revised paper to ensure they are satisfied, then I'll check and decide if we can accept at that point. Thanks for submitting to PeerJ!

Reviewer 1 ·

Basic reporting

Please see general comments

Experimental design

Please see general comments

Validity of the findings

Please see general comments

Additional comments

Herrel et al. have submitted a study attempting to link jump performance (distance and endurance) to morphology. They found, for example, evidence for a force-endurance tradeoff in females, but not in males. Their conclusion is a list several comparisons among many parameters, making it difficult for the reader to extract a central finding. However, an interesting claim is made that selective pressures are sex-specific, but it is not well supported or explained. Additionally, I have general technical concerns with data collection and analysis that must be addressed before the ms is suitable for publication.

General comments:
1. My main concern is that although many of the relationships reported are statistically significant, the r^2 values are all very low, indicating unexplained variability. The authors have not addressed this unexplained variability. The authors have used partial regression statistics in part of the analysis - however, I suggest a more thorough partial regression analysis to account for additional variability. For example, although it’s intriguing that ilium length correlates to force in males, it only explains less than 29% of the variation in peak force. Perhaps there is something meaningful here, but the authors do not report evidence to rule out other possibilities. For example, maybe ilium length correlates to some other morphological feature (neglected from the analysis) that more directly contributes to force?

2. Another large concern is that the authors did not mention whether they elicited maximum performance for each frog. If each individual is hopping at some fraction of its maximum capacity, there is a large random parameter in the data set which may contribute to the large variability observed. If this is the case, how can individuals be compared to one another? Thus, I do not see how the authors can make their claim if the locomotor motivation varies among individuals.

3. From the figure, medio-lateral forces seem significant, however they are absent from the analysis. These forces should be explained – are the frogs producing these high lateral forces because they are forced to jump on a circular track (i.e. always turning)? Or is this a random variable that is influencing the resultant force, but not horizontal/vertical components (thus biasing the results)?

4. Throughout the ms, there is a focus on peak ground reaction forces compared to limb length. However, shouldn’t limb length correlate more strongly to the duration of force on the ground? If so, I feel the authors should measure impulse instead of (or in addition to) peak forces, especially since peak forces for the smallest frogs are prone to greater error (lower signal/noise) in the force recordings. Using impulse measurements might reduce the unexplained variability in the data set.

5. One central claim seems to be that selection acts differentially on males vs. females. But, this claim is presented as an afterthought at the end of the discussion and the arguments preceding are not compelling. The authors should better explain why their data lead to this conclusion. Additionally, one piece of evidence for differential selection comes from Fig. 3 showing that males produce greater force at a given size. However, the different body size distributions between sexes is not mentioned in the paper. It seems that selection is acting on body size itself such that absolute forces are nearly equal between sexes. In other words, males may be smaller, on average compared to females (from inspecting fig. 3 by eye). Regardless, the authors should report information on the size distribution of males vs. females to see if this influences the data are interpreted.

Specific comments:
1. L88-89: Females invest more of what, specifically (calories, etc.)?

2. Methods: How was jump distance measured? Were individual jump distances measured? Was this done with high speed video?

3. L166-174: Resultant and vertical forces are compared throughout, but horizontal forces are only mentioned in the first comparison and medio-lateral forces are neglected entirely which seems strange given that they are similar in magnitude as horizontal (e.g. Fig. 1).

4. L192-205: I’m confused why r values are reported in this section whereas r^2 values are reported in the previous section.

5. L166-205: Many comparisons are made here, making it difficult to keep track. The authors should consider repeating these correlation data in a table to make the results easier to digest and compare.

6. Fig. 1: It seems confusing and odd to include the entire time trace, including the time before the animal is on the plate. I.e. it’s unnecessary and confusing to show the force response to the animal being placed on the plate (before 7.5 s). I suggest truncate the plot so it begins at ~7.5 s and ends ~8 s so that the details of the actual jump are expanded. Also, why isn’t the left-right force mentioned in the legend?

7. Fig. 2 and following: What do the individual points on the scatter plots represent? Are they individual jumps or a single maximum value for all jumps of any given individual?

·

Basic reporting

Line 78 to 82: I’m not sure enough information is given to fully follow the argument underpinning this hypothesis. E.g. variation in muscle histochemistry could theoretically cause a trade-off between the performance of a single jump and the distance jumped to exhaustion. It would be useful to have the regression equations stated in the figure legends.

Experimental design

No comments

Validity of the findings

Did the authors test for repeatability in jumping and swimming performance? It would be useful to know how repeatable these measures were in this study or for the previously determined repeatability of these measures in other studies to be cited.

In the stepwise multiple linear regression analysis the first model produced will reflect the anatomical variable that best predicts the variation in the performance variable. However, there may be other anatomical variables that are also highly correlated, maybe almost as highly correlated, with that performance variable. I think that the way your data is presented hides some of this information. There may be correlation between the anatomical variables as well and, dependent on the degree of such correlation (assessment of variable inflation factor), other authors might have opted to use principal component analysis prior to regression analysis. I suggest that the correlations between anatomical variables and performance data is presented, as well as presentation of the multiple regression data, and/or the complete data set is made available and/or analysis of VIF values is undertaken to determine whether colinearity of anatomical variables is an issue.

Additional comments

This is an interesting manuscript demonstrating that: males produced higher jump forces, than females, relative to body mass or leg length; In each sex different anatomical variables predicted variation in jump force; Peak jump forces and distance to exhaustion were all correlated with eachother, whereas peak horizontal force was correlated to maximal swimming performance, but that these relationships differed between sexes.

There are a few typos in the manuscript e.g. Line 218 replace “while” with “why”. Line 259 replace “t” with “it”. Please have another check throughout.

Figures 3 and 4: please state in the figure legends what the open symbols and closed symbols refer to. Shouldn’t Figure 4B also have a regression line on it?

---

## Round 0.2 · Minor Revisions

· Academic Editor

Minor Revisions

The reviewers agree that the study is almost publishable and make some final helpful suggestions for polishing up the paper. Please send a new Response document with all points addressed and I will check that- if satisfied I'll swiftly hop to accept the paper!

Reviewer 1 ·

Basic reporting

Please see general comments

Experimental design

Please see general comments

Validity of the findings

Please see general comments

Additional comments

The revised manuscript is improved and is now close to being ready for publication. I appreciate the additional table data as well as the discussion speculating the source of unexplained variation. Both are very helpful. Also, the authors have filled in the crucial missing gaps in their Methods. Given these gaps, it should not have been surprising to the authors that I was confused. Yet, the tone of the authors' response seemed to suggest that I had overlooked something obvious - this was not the case.

I have minor points, all easy to address:

I’m not convinced that the authors have elicited maximum performance in this study. This requires clearer wording. Instead of “This typically elicited escape responses from the individuals causing them to jump as far as possible away from the observer..” I suggest “To elicit maximal performance for each individual, frogs were placed on the force plate….”. Please delete the word ‘typically’ as it implies that maximal performance was only sometimes elicited, questioning the validity of the study. I.e. how can we believe that the morphology-performance regressions are meaningful if the authors didn’t elicit maximum performance?

Fig. 2 and elsewhere, please clarify that these scatterplot points are for single maximal jumps for each given individual. The authors have clarified this in Fig. 4, but not the other scatterplots.

Finally, for the record (and perhaps not for consideration with the current ms), I am not convinced that summing medio-lateral force with fore-aft force is appropriate here given that medio-lateral force (labeled in fig 1 as left-right) is equal in magnitude to fore-aft. The vector sum of fore-aft + medio-lateral will influence the net magnitude, however, the medio-lateral presumably shouldn’t contribute to jump distance unless the animals are turning. Thus, variability in medio-lateral force will add additional unexplained variation to the summed horizontal force. Forces should therefore be treated independently, in my view. So, I think the authors may be overlooking aspects of their data that would shed light on their unexplained variability. However, since the authors admit that they are more concerned with ‘peak performance’ than with ‘mechanics of jumping’, perhaps a more detailed analysis of force components is better left for a future study.

·

Basic reporting

no comments

Experimental design

no comments

Validity of the findings

no comments

Additional comments

I think that this manuscript is almost ready for publication. Inclusion of the tables of correlations makes for interesting reading, further strengthening some of the discussion. The combined data and the female data really demonstrates that larger size is correlated with increased jump distance and that there is little difference in r value between many morphological traits. However, the male data clearly shows that variation in ileum size is the key correlate with jumping performance; very interesting.

I have no further major comments, but have a few minor suggestions below for the authors consideration.

Line 130 I suggest changing “recoding” to “recording”
Lines 134-135 Why not also cite the previous papers that you mention in “our reply” that further demonstrate repeatability in locomotor traits.

Line 236 Is it worth pointing out that these are differences between sex?
Line 277 I suggest changing “higher performance” to “higher locomotor performance”
Line 278 I suggest changing “see” to “seek”

Table 2 legend. I recommend changing “ms” to “m.s” to avoid possible confusion with milliseconds.

Figure legends: I think that the figure legends at the end of the manuscript are the correct ones and that the figure legends on the same pages as the figures are in some cases older less detailed versions. It would be useful to include the equations of the regression lines and r2 values in the figure legends.

---

## Round 0.3 · accepted · Accept

· Academic Editor

Accept

I'm satisfied that these revisions are sufficient, so I am happy to accept this paper now- congrats!